# Reliability of left atrial strain reference values: A 3D echocardiographic study

**Yosuke Nabeshima** [1] *, **Tetsuji Kitano** [2], **Masaaki Takeuchi** [3]

**1** Second Department of Internal Medicine, University of Occupational and Environmental Health, School of Medicine, Kitakyushu, Japan, **2** Department of Cardiology and Nephrology, Wakamatsu Hospital of the University of Occupational and Environmental Health, Kitakyushu, Japan, **3** Department of Laboratory and Transfusion Medicine, University of Occupational and Environmental Health Hospital, Kitakyushu, Japan

* y.nabeshima1016@gmail.com

**Data Availability Statement:** All relevant data are within the paper and its Supporting information files.

**Funding:** The authors received no specific funding for this work.

## Abstract

### Background

Standard apical four-chamber and two-chamber views often maximize the long-axis of the left ventricle, resulting in artifactitious foreshortening of the left atrium (LA), which may overestimate LA longitudinal reservoir strain (LALS). We compared LALS values between 2D echocardiography (2DE) and 3D echocardiography (3DE) in healthy subjects to determine whether 2DE speckle tracking analysis overestimates the reference value of LALS.

### Methods and results

In this study, 4 types of cohorts were included: 1. 105 normal subjects (retrospectively), 2. 53 patients with cardiovascular diseases (retrospectively), 3. 15 patients who received cardiac magnetic resonance (prospectively), and 4. 20 normal subjects (prospectively). LALS and LA length were measured using both 2DE and 3DE in 105 healthy subjects (median age: 42 years). Biplane LALS was measured in apical four- and two-chamber views using 2DE speckle tracking software, and 3DE LALS was measured using new 3DE LA strain software. To determine sensitivity, we also performed the same analysis in 53 patients with cardiovascular disease. The mean value of biplane LALS was 39.6%. LA length at both end-diastole (r = -0.43) and end-systole (r = -0.54) was negatively correlated with biplane LALS. Multivariate regression analysis revealed that both end-diastolic and end-systolic LA length had significant negative relationships with biplane LALS after adjusting for anthropometric and echocardiographic image quality parameters. 3DE LALS (23.7±7.6%) gave significantly lower values than 2DE LALS (39.5±12.0%, p<0.001) with a weak correlation (r = 0.33). LA length measured by 2DE was significantly shorter than that measured by 3DE. The same trend was observed in diseased patients.

### Conclusions

Our results revealed that in 2DE, the LA cavity consistently appears longitudinally foreshortened in apical views, potentially overestimating LALS. 3DE may overcome this limitation.

**Competing interests:** The authors have declared that no competing interests exist.

## Introduction

The left atrium is responsible for left ventricular (LV) diastolic filling; thus, its volumetric and mechanical changes reflect the degree and chronicity of LV diastolic dysfunction [1, 2]. Left atrial (LA) longitudinal strain, assessed by two-dimensional echocardiography (2DE) speckle tracking analysis, has become increasingly common for estimating LV diastolic dysfunction [3–5] and for predicting outcomes [6, 7]. In a recent meta-analysis, a normal value of 39% was reported for LA reservoir strain [8]. The standardization task force of the European Association of Cardiovascular Imaging and the American Society of Echocardiography recommends that "LA strain should be measured using a non-foreshortened apical four-chamber view of the left atrium" [9]. Since an apical four-chamber view is usually intended to fully delineate the long-axis diameter of the left ventricle, and because the long axes of the left ventricle and the left atrium do not lie in the same 2D cutting plane [10, 11], this approach risks an artificial foreshortening of the left atrium. Voigt et al. recently published a methodological paper explaining how to measure 2D LA strain [12], in which they presented a case indicating that a foreshortened LA image results in higher LA strain. However, there was no evidence that supports their recommendation. We hypothesized that a foreshortened LA view could overestimate 2DE LA strain because it reduces the initial perimeter of the region of interest (ROI), which is the denominator of the strain calculation. Recently, novel software to measure LA strain using 3D echocardiography (3DE) datasets has been developed.

Accordingly, in this study we sought to (1) determine the impact of LA length on 2DE LA longitudinal strain values, (2) to compare LA longitudinal strain assessed by 2DE and 3DE, and (3) to compare LA length determined with 2DE and 3DE.

## Methods

### Study population

We retrospectively selected 105 healthy adult volunteers (≥20 years old) who had undergone both 2DE and 3DE examinations using ultrasound equipment from a single manufacturer (GE Healthcare, Horten, Norway) from our 3DE database to determine reference values for 2DE and 3DE LA longitudinal strain. To determine whether results are consistently observed in patients, we also selected from the database, 53 adult patients with cardiovascular disease who underwent both clinical 2DE and 3DE examinations. For the sensitivity analyses, we prospectively collected 15 adult patients who were clinically indicated for cardiac magnetic resonance and 20 healthy adult volunteers. The ethics committee in the University of Occupational and Environmental Health approved the study protocol. The need to obtain informed consent was waived for the retrospective cohorts. Informed consent was obtained from prospective cohorts.

### Echocardiographic acquisition

All subjects underwent comprehensive 2DE, 3DE, and Doppler echocardiography examination with a commercially available ultrasound machine and transducer (E95 with an M5S probe for 2DE and a 4V or 4Vc probe for 3DE, GE Healthcare, Horten, Norway). We carefully acquired 2DE apical views to show long axis diameters from the LV apex to the LA roof for as much of the LA and LV lengths as possible. Multi-beat, full-volume 3DE datasets were also acquired using an apical approach. Depth and sector angles were adjusted to include the entire left ventricle and left atrium with the highest volume rate.

## 2DE analysis

The LA wall in 2D apical four-chamber and two-chamber views was divided into six segments, and image quality of each segment was scored as 0 (no visualization of the LA endocardium), 0.5 (partially visualized), or 1 (full visualization), and summed to yield a total image score. Image quality was graded as either 'good' (defined as a total score > 10), 'fair' (8 < score ≤ 10), 'poor' (6 < score ≤ 8) or 'extremely poor' (score ≤ 6). The LA endocardial border was traced at the end-diastolic and end-systolic frames on the apical four-chamber and two-chamber views, from which LA length (from the center of the connecting line between both sides of the mitral annulus to the LA roof) was measured (Fig 1). Pulmonary veins and left atrial appendages were carefully excluded from the endocardial border. Values were averaged to obtain mean maximal and minimal 2DE LA lengths. LA volume was determined using Simpson's biplane method. LA emptying fraction (LAEF) was calculated as ([maximal LA volume —minimal LA volume] / maximal LA volume) x 100.

Using the same 2DE image clip, the ROI of the LA wall was determined at the end-systolic frame. Width of the ROI was set as small as possible. LA longitudinal reservoir strain was measured using 2DE speckle tracking software (2D Strain, EchoPAC PC version 203, GE Healthcare). We used a zero-strain reference set at LV end-diastole, i.e., R-R gating [9]. The software divided the LA wall into six segments, and generated six segmental LA longitudinal strain curves. Using the averaged LA longitudinal strain curve from the six segments, we determined the LA longitudinal reservoir strain, reflecting peak values of LA strain in each view. Since LA longitudinal reservoir strain using 3DE was derived from the LA endocardial border, we used endocardial LA longitudinal strain values derived from layer-specific strain analysis for the comparison. Biplane LA longitudinal strain was calculated by averaging the four-chamber and two-chamber endocardial LA longitudinal strains. We determined the feasibility of 2DE speckle tracking software according to the quality of tracking after manual endocardial tracing.

Other conventional echocardiographic parameters were derived as described below. LV volumes and LVEF were measured using modified biplane Simpson's method. LV inflow velocity was recorded at the tip of the mitral leaflet using pulsed-wave Doppler echocardiography, and mitral annular velocity at both sides of the mitral annulus was recorded with tissue Doppler echocardiography.

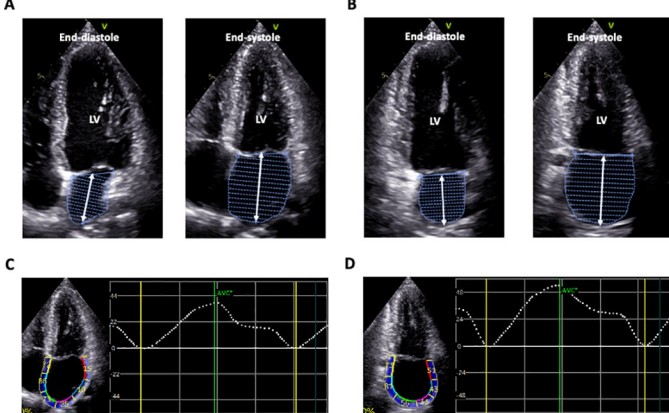

**Fig 1. How to measure left atrial volume, length, and strain using 2D echocardiography.** A and B: left atrial endocardial border tracking on the apical four-chamber (A) and two-chamber (B) views from which disk-area and length were determined. C and D: Corresponding left atrial strain curves.

## 3DE analysis

3DE image quality was assessed in 2D apical four-chamber and two-chamber views extracted from 3DE datasets using the same criteria as in 2DE. 3DE LA strain analysis was performed using newly developed software (4D Auto LAQ, GE Healthcare). At first, the center of the mitral valve was determined and the angle was adjusted to show the full delineation of long-axis LA diameter in apical four-chamber, two-chamber, and long-axis views extracted from 3DE datasets (Fig 2A). Next, the software automatically determined the LA endocardial border in 3D space throughout a single cardiac cycle, using an extended Kalman filter that combines LA geometry, a motion model, and edge detection algorithms (Fig 2B). The endocardial border was manually adjusted at end-diastolic, end-systolic, and pre-atrial contraction frames, as required. As addition to 2DE analysis, pulmonary veins and left atrial appendages were carefully excluded from the endocardial border. Subsequently, the software generated the LA volume and LA longitudinal strain curves, from which we measured maximal LA volume, minimal LA volume, LAEF, and LA strain (Fig 2C). In this study, both 2D and 3D strain analysis software employed the same formula to compute the change in length (L) of the endocardial border = (L(end-systole)–L(end-diastole)) / L(end-diastole) x 100%. We determined the feasibility of 3DE speckle tracking software according to the possibility of adequate tracking after manual correction of automatically detected endocardial border as required.

To measure the 3D LA long-axis diameter, anterior-posterior and commissure-commissure views showing full delineation of the LA long axis diameter were extracted from 3DE datasets (Fig 2D) under the guidance of the LA short axis view [13]. From both views, we measured the long-axis diameter of the left atrium between the center of the mitral annular plane and the LA roof at end-diastole and end-systole (Fig 2E and 2F). These values were averaged to determine the mean 3DE-determined maximal and minimal LA lengths.

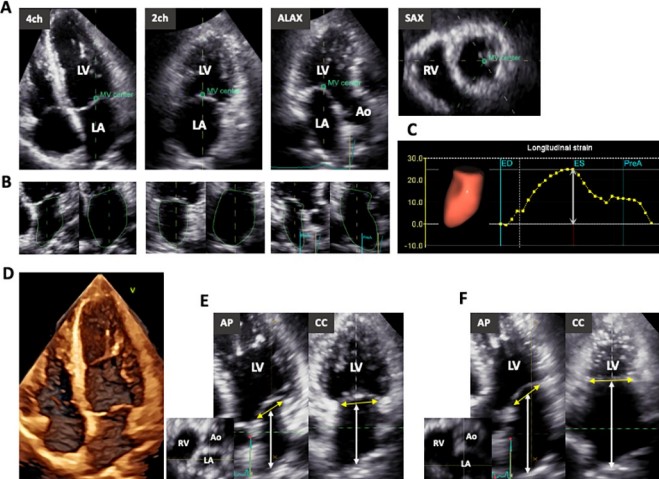

**Fig 2. How to measure left atrial strain and length using 3D echocardiography.** A: The center of the mitral valve was registered and the angle was changed so that dotted lines dissected the maximal LA length in three apical views extracted from 3DE datasets. B: Software determined the LA endocardial border in 3D space. C: LA longitudinal strain curve and a 3D cast of the left atrium. D: Cropped view of 3DE datasets. E and F: Under the guidance of the short axis view of the left atrium, anterior-posterior and commissure-commissure views showing full visualization of LA longitudinal distance were obtained, from which we measured the LA longitudinal distance between the center of the mitral annular plane and the LA roof at end-diastole and end-systole. 3DE, three-dimensional echocardiography; Ao, aorta; AP, anterior-posterior view; CC, commissure-commissure view; ED, end-diastole; ES, end-systole; MV, mitral valve; LA, left atrium; LV, left ventricle; RV, right ventricle; PreA, pre-atrial contraction.

## Measurement variabilities

Intra-observer variability was assessed by having the observer repeat the measurement of both 2DE biplane LA longitudinal strain and 3DE maximal and minimal LA volumes, LAEF, and LA longitudinal strain at two-week intervals in 15 randomly selected, healthy subjects. Inter-observer variability was determined by employing a second observer to perform these measurements in the same 15 subjects. Intra- and inter-observer variability values were calculated as absolute differences between the two corresponding measurements as percentages of their mean and intraclass correlation (ICC).

## Validation study using cardiac magnetic resonance

To validate the relationship between the LA foreshortened view and LA strain analysis, we prospectively acquired apical four-chamber and two-chamber, steady-state free precession (SSFP), dynamic gradient-echo cine loops optimized for the left ventricle, and those optimized for the left atrium in 15 patients who had clinically indicated cardiac magnetic resonance (CMR) examinations. CMR imaging was performed with a 3.0T scanner (SIGNA Premier, GE Healthcare Milwaukee, WI) with a phased-array cardiovascular coil. On apical long-axis SSFP images, the optimal cutting plane was individually determined to visualize LV (or LA) long-axis as long as possible (Fig 3). These planes were rotated to the same degree to obtain apical four-chamber and two-chamber views. LA strain was measured using feature tracking software (2DCPA MR, TomTec Imaging Systems GmbH, Unterschleissheim, Germany). The software also provided maximal and minimal LA volumes. Values for apical four-chamber and two-chamber views were averaged and compared between LV-focused views and LA-focused views. We also compared the LAVs and LALS between the CMR biplane method and the 2DE biplane method.

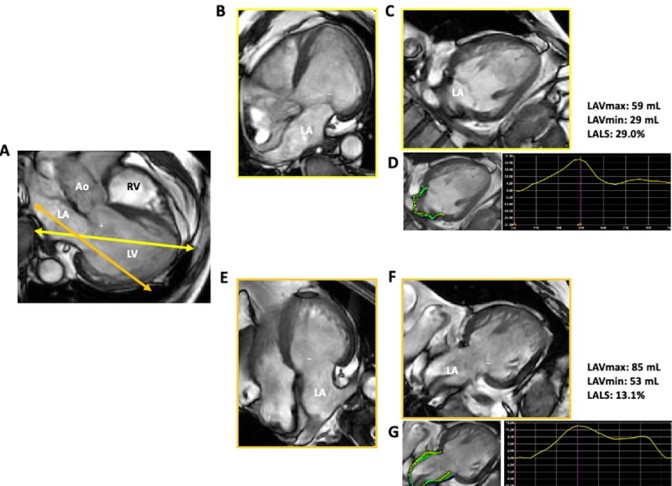

**Fig 3. CMR feature tracking analysis of LA.** A: A cut-plane optimized for the left ventricle (yellow arrows) and that optimized for the left atrium (orange arrows) were determined using apical long-axis SSFP images. These planes rotated to the same degree to generate apical four-chamber and two-chamber views aimed for the left ventricle (B and C) and those aimed for the left atrium (E and F). D: Feature tracking of the LA wall using LV-focused, two-chamber view and the corresponding LA strain curve. G: Feature tracking of the LA wall using LA-focused, two-chamber view and the corresponding LA strain curve. LAVmax, maximal LA volume; LAVmin, minimal LA volume; LALS, LA longitudinal strain.

## Acquisition of non-foreshortened LA images

Finally, we prospectively acquired apical four-chamber and two-chamber views aimed for non-foreshortened LA in 20 healthy subjects (mean age: 31 ± 6 years, 16 men). LA long-axis diameter and LA strain values were averaged, and compared to the corresponding values measured from standard apical four-chamber and two-chamber views and 3DE.

## Statistical analysis

Continuous data were expressed as means ± SD or median and interquartile range (IQR; 25th percentile—75th percentile). Categorical data were presented as absolute numbers or percentages. The Wilcoxon matched-pairs signed rank test or the Friedman test was used to evaluate paired comparisons between the two groups or among the three groups. Post-hoc comparisons were performed using Dunn's test. A Pearson correlation analysis was performed between pairs of continuous variables. Univariate and multivariate linear regression analysis were conducted to determine the association between anthropometric and echocardiographic variables and 2DE or 3DE LA longitudinal strain measurements. Variables showing $p<0.1$ were used for multivariate regression analysis with stepwise selection based on Akaike's Information Criterion. Collinearity was considered present at $r>0.70$. A two-sided p-value $<0.05$ was considered statistically significant. All statistical analyses were performed using commercial software (SPSS version 24, Chicago, IL; R version 3.4.3, The R Foundation for Statistical Computing, Vienna, Austria).

## Results

### Study subjects

A total of 105 healthy subjects were selected for the analysis (Table 1). 2DE image quality was 'good' in 34 subjects (32%), 'fair' in 42 subjects (40%), 'poor' in 24 subjects (23%) and

**Table 1. Clinical and echocardiographic characteristics in 105 normal subjects.**

| Variable | Mean ± SD or Median (p25 to p75) |
|---|---|
| Age (year) | 42 (30 to 56) |
| Male/female | 59/46 |
| Height (cm) | 167 (157 to 173) |
| Weight (kg) | 63 (54 to 72) |
| Body surface area (/m$^2$) | 1.70 (1.53 to 1.84) |
| Heart rate (bpm) | 65 ± 10 |
| Systolic blood pressure (mmHg) | 129 ± 10 |
| Diastolic blood pressure (mmHg) | 75 ± 9 |
| 2DE LV end-diastolic volume (mL) | 106 ± 25 |
| 2DE LV end-systolic volume (mL) | 48 ± 13 |
| 2DE LVEF (%) | 55 ± 4 |
| E wave velocity (cm/sec) | 79 ± 17 |
| A wave velocity (cm/sec) | 57 ± 17 |
| E/A | 1.45 (1.07 to 1.79) |
| s' average (cm/sec) | 9.7 ± 1.9 |
| e' average (cm/sec) | 12.9 ± 3.7 |
| E/e' average | 6.05 (5.11 to 7.36) |

2DE, 2-dimensional echocardiography; e', early diastolic peak mitral annular velocity; LVEF, left ventricular ejection fraction; p25, 25th percentile; p75, 75th percentile; s', peak systolic mitral annular velocity.

e' and s' represent averaged values measured on both side of the mitral annulus.

'extremely poor' in 5 subjects (5%). Corresponding values of 3DE were 35 (33%), 35 (33%), 24 (23%), and 11 (10%), respectively. Median values for 2DE frame rate and 3DE volume rate were 61/sec and 26/sec, respectively. 2DE analysis was possible in all subjects. 3DE analysis was not possible in 11 subjects due to erroneous LA contours generated by the software that could not be adequately edited; hence, feasibility was 90%. The median value of the image quality score in patients whose 3DE analysis was not possible (5.5, IQR; 5 to 6.25) was significantly lower than that of those on whom 3DE analysis could be performed (9.5, IQR; 8 to 10.5, p<0.001).

## 2DE analysis

Paired comparisons demonstrated that two-chamber LA longitudinal strain (42.4±15.1%) was significantly larger than four-chamber LA longitudinal strain (36.8±12.2%, p<0.001, Fig 4A). The biplane LA longitudinal strain was 39.6±11.8%. LA length measured using the apical two-chamber view was significantly shorter than that measured using the apical four-chamber view at both end-diastole and end-systole (Fig 4B). There were significant inverse correlations between biplane LA longitudinal strain and average LA length at end-diastole (r = 0.54, p<0.001, Fig 4C) and between biplane LA longitudinal strain and average LA length at end-systole (r = 0.43, p<0.001, Fig 4D).

Table 2 depicts univariate and multivariate linear regression analyses for the association of anthropometric and echocardiographic parameters with 2DE LA longitudinal strain. Because there was collinearity between the minimal and maximal LA lengths, we performed two types

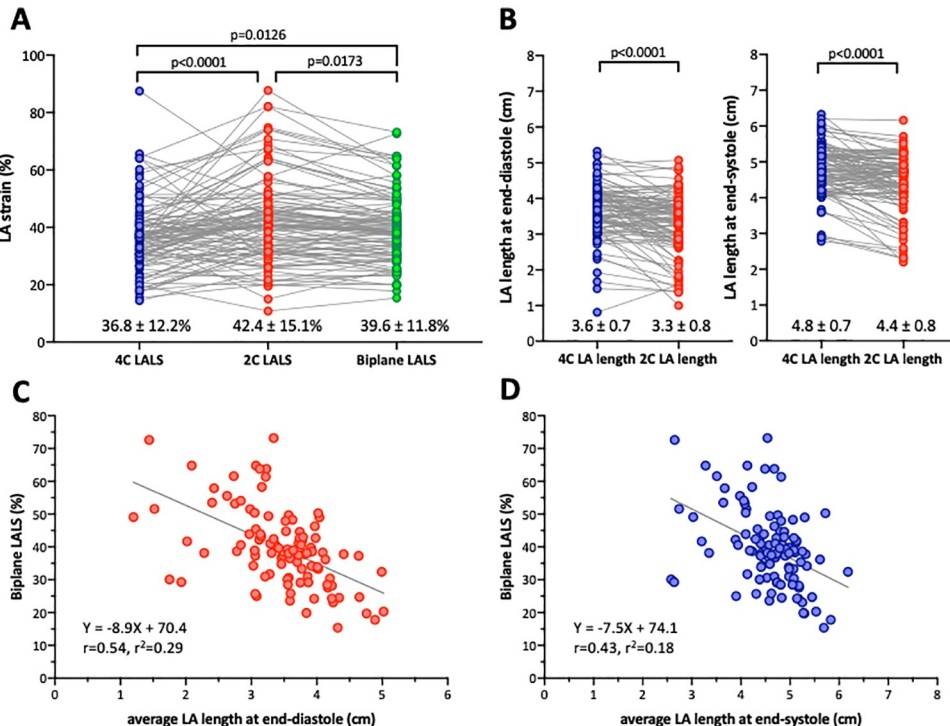

**Fig 4. Comparisons of left atrial strain and length using 2DE analysis.** A: LA longitudinal strain in apical four-chamber view (4C LALS), apical two-chamber view (2C LALS), and their average (biplane LALS). B: LA length comparing apical four-chamber and apical two-chamber at end-diastole (left panel) and at end-systole (right panel). C: A linear correlation between average left atrial length at end-diastole and biplane left atrial longitudinal strain. D: A linear correlation between average left atrial length at end-systole and biplane left atrial longitudinal strain.

**Table 2. Univariate and multivariate linear regression analysis for the association of anthropometric and echocardiographic parameters with left atrial longitudinal strain assessed with 2D echocardiography in 105 normal subjects.**

| | 4C LALS | | | | | | 2C LALS | | | | | | Biplane LALS | | | | | |
|---|---|---|---|---|---|---|---|---|---|---|---|---|---|---|---|---|---|---|
| | Univariable | | Multivariable 1* | | Multivariable 2† | | Univariable | | Multivariable 1* | | Multivariable 2† | | Univariable | | Multivariable 1* | | Multivariable 2† | |
| | t | p | t | p | t | p | t | p | t | p | t | p | t | p | t | p | t | p |
| Age | -3.84 | <0.001 | -2.66 | 0.009 | -2.19 | 0.030 | -3.89 | <0.001 | | | | | -4.59 | <0.001 | | | | |
| Male | -2.33 | 0.022 | | | | | -1.73 | 0.085 | | | | | -2.32 | 0.021 | | | | |
| BSA | -2.73 | 0.007 | | | | | -2.50 | 0.013 | | | | | -3.05 | 0.002 | | | | |
| HR | 1.75 | 0.081 | | | | | 2.24 | 0.027 | | | | | 2.35 | 0.020 | | | | |
| SBP | -1.54 | 0.125 | | | | | -2.81 | 0.005 | | | | | -2.62 | 0.010 | | | | |
| DBP | -2.46 | 0.015 | | | | | -2.86 | 0.005 | | | | | -3.15 | 0.002 | | | | |
| IQ good | 1.39 | 0.168 | | | | | 2.87 | 0.005 | 2.51 | 0.013 | 2.51 | 0.013 | 2.54 | 0.012 | 2.43 | 0.016 | 2.47 | 0.015 |
| IQ fair | 1.09 | 0.280 | | | | | 3.21 | 0.001 | 2.81 | 0.005 | 2.68 | 0.008 | 2.60 | 0.010 | 2.34 | 0.021 | 2.21 | 0.028 |
| IQ poor | 1.15 | 0.254 | | | | | 2.96 | 0.003 | 1.86 | 0.065 | 1.56 | 0.121 | 2.48 | 0.014 | 1.41 | 0.160 | 1.17 | 0.242 |
| FR | -1.61 | 0.111 | | | | | -0.51 | 0.606 | | | | | -1.15 | 0.249 | | | | |
| LA length max | -4.38 | <0.001 | -2.87 | 0.005 | | | -5.05 | <0.001 | -2.85 | 0.005 | | | -4.87 | <0.001 | -2.70 | 0.008 | | |
| LA length min | -5.29 | <0.001 | | | -3.48 | <0.001 | -6.72 | <0.001 | | | -4.24 | <0.001 | -6.52 | <0.001 | | | -3.96 | <0.001 |

Adjusted $R^2$ = 0.22, Adjusted $R^2$ = 0.25 Adjusted $R^2$ = 0.30, Adjusted $R^2$ = 0.36 Adjusted $R^2$ = 0.33, Adjusted $R^2$ = 0.38

* including LA length max.

† including LA length min.

of multivariate linear regression analysis, including minimal or maximal LA length. There was a significant negative correlation between LA length and four-chamber LA longitudinal strain. Multivariate linear regression analysis revealed that LA lengths at both end-diastole and end-systole had a significant negative association with four-chamber LA longitudinal strain after adjusting for age, gender, body surface area, and diastolic blood pressure. Image quality was also correlated with two-chamber LA longitudinal strain. Both LA length at end-diastole and end-systole had significant negative associations with two-chamber LA longitudinal strain after adjusting for anthropometric factors and image quality. The same trend was also observed for biplane LA longitudinal strain.

## Comparisons of 2DE and 3DE analyses

LA longitudinal strain assessed by 3DE was 23.7±7.6%. Fig 5 presents pairwise comparisons of LA volumes, LAEF, and LA longitudinal strain in 94 subjects for whom both 2DE and 3DE LA strain analyses were possible. Although there were no significant differences in maximal and minimal LA volumes and LAEF between 2DE and 3DE, LA longitudinal strain using 3DE was significantly lower than the value derived using 2DE. LAEF and LA longitudinal strain measurements from the two methods were poorly correlated.

A paired comparison of 2DE and 3DE LA lengths revealed that LA lengths determined with 2DE at both end-diastole (p<0.001) and end-systole (p<0.001) were significantly shorter than values obtained using 3DE (Fig 6).

The absolute difference between 2DE and 3DE LA strain was 15.7±12.0%, which was modestly correlated with differences in LA lengths between 2DE and 3DE at both end-diastole (r = 0.29, p = 0.004) and end-systole (r = 0.31, p<0.003). Multivariate linear regression analysis revealed that differences in LA length at both end-diastole (p = 0.002) and end-systole (p = 0.001) had a significant association with differences in LA longitudinal strain after adjusting for age, sex, systolic blood pressure, and volume rate.

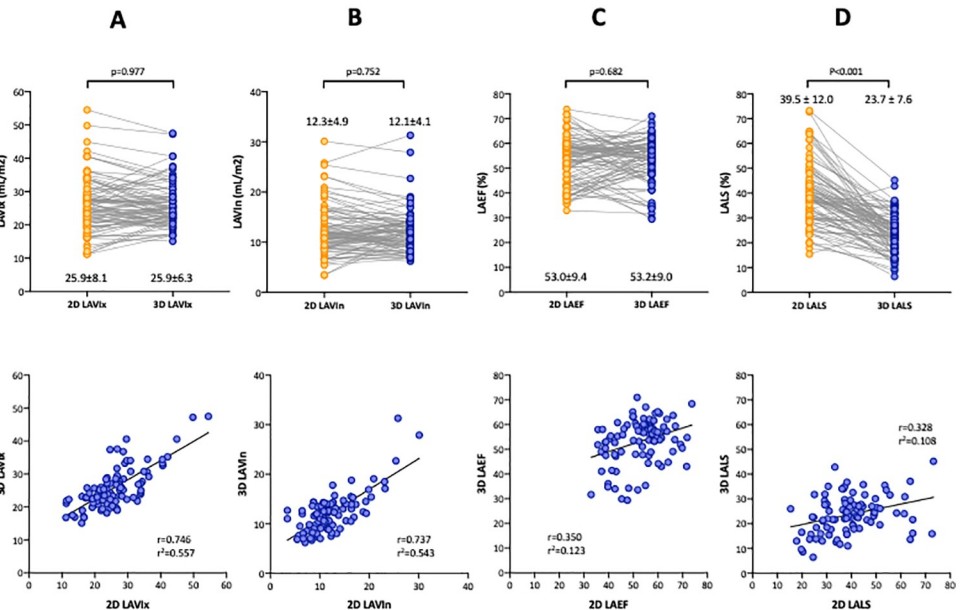

**Fig 5. A paired comparison and correlation of left atrial volumes, emptying fraction, and strain between 2DE and 3DE.** A and B: Indexed maximal and minimal left atrial volumes (LAVIx, LAVIn) between 2DE and 3DE (upper panel) and their correlation (lower panel). C: Left atrial emptying fraction (LAEF) between the two methods (upper panel) and their correlation (lower panel). D: Left atrial strain between the two methods (upper panel) and their correlation (lower panel).

The difference in LA lengths between 2DE and 3DE at both end-diastole (r = 0.21, p<0.001) and end-systole (r = 0.30, p<0.001) was modestly correlated with 2DE LA strain. Multivariate linear regression analysis revealed that differences in LA length at both end-diastole (p<0.001) and end-systole (p = 0.002) had a significant association with 2DE LA strain after adjusting for age, sex, systolic blood pressure, volume rate, 3DE image quality, and minimal or maximal 3DE LAV.

## 3DE analysis

Univariate analysis showed that age, diastolic blood pressure, and LA length were negatively associated with 3DE LA longitudinal strain (Table 3). Because of collinearity between the

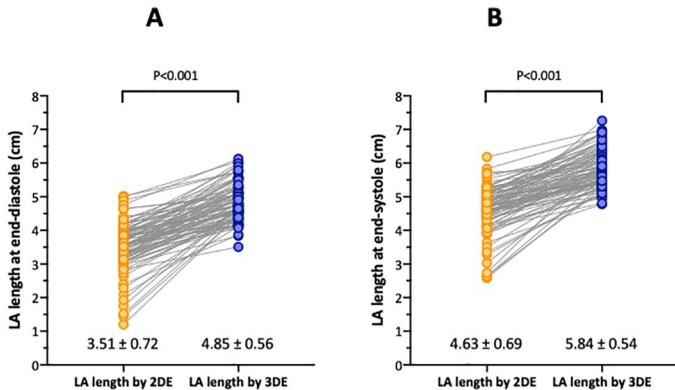

**Fig 6. A paired comparison of left atrial length between 2DE and 3DE.** A: End-diastole. B: End-systole.

**Table 3. Univariable and multivariable linear regression analysis for the association of left atrial longitudinal strain assessed with 3D echocardiography in 94 normal subjects.**

| | 3D LA longitudinal strain | | | | | |
| | Univariable | | Multivariable 1* | | Multivariable 2† | |
| | t-value | p-value | t-value | p-value | t-value | p-value |
|---|---|---|---|---|---|---|
| Age | -5.044 | <0.0001 | -3.780 | <0.0001 | -3.141 | <0.0001 |
| Sex (male) | -1.473 | 0.1439 | | | | |
| BSA | 0.033 | 0.9729 | | | | |
| HR | 0.281 | 0.7785 | | | | |
| SBP | -1.641 | 0.1040 | | | | |
| DBP | -2.329 | 0.0220 | | | | |
| IQ good | 1.414 | 0.1605 | | | | |
| IQ fair | 1.294 | 0.1987 | | | | |
| IQ poor | 1.162 | 0.2480 | | | | |
| VR | 0.887 | 0.3370 | | | | |
| Max. LA length | -2.501 | 0.0141 | | | | |
| Min. LA length | -4.357 | <0.0001 | | | -2.298 | 0.0239 |

Adjusted $R^2$ = 0.198, p<0.0001 Adjusted $R^2$ = 0.234, p<0.0001

VR, volume rate. Other abbreviations were the same as Table 1.

* including LA length max.

† including LA length min.

minimal and maximal LA lengths, we performed two types of multivariate linear regression analyses, including minimal or maximal LA length. Multivariate linear regression analysis revealed that 3DE determined end-systolic LA length was no longer significantly associated with 3DE LA longitudinal strain after adjusting for age and diastolic blood pressure (Table 3). 3DE-determined, end-diastolic LA length was still significantly associated with 3DE LA longitudinal strain after adjusting for age and diastolic blood pressure. However, the association was weak compared with 2DE analysis.

## Sensitivity analysis

Table 4 depicts clinical and echocardiographic characteristics of 53 patients with cardiovascular disease. 2DE image quality was 'good' in 12 subjects (23%), 'fair' in 24 subjects (45%), 'poor' in 14 subjects (26%) and 'extremely poor' in 3 subjects (6%). Corresponding values of 3DE image quality were 15 (28%), 21 (40%), 14 (26%), and 3 (6%), respectively. The feasibility of 2DE and 3DE analyses were 100% and 94%, respectively. There were significant inverse correlations between biplane LA longitudinal strain and average LA length at end-diastole (r = 0.65, p<0.001, Fig 7A) and between biplane LA longitudinal strain and average LA length at end-systole (r = 0.55, p<0.001, Fig 7B). LA longitudinal strain using 3DE (13.2 ± 6.8%) was significantly lower than values derived using 2DE (20.6 ± 9.2%, p<0.001) and the two values were moderately correlated (r = 0.77). 2DE-determined LA length was significantly shorter than 3DE-determined LA length at both end-diastole (5.64 ± 0.77 mm vs. 6.49 ± 0.82 mm, p<0.001) and end-systole (4.81 ± 0.88 mm vs. 5.85 ± 0.95 mm, p<0.001). Because there was collinearity between the minimal and maximal LA lengths, we performed two types of multivariate linear regression analyses including minimal or maximal LA length. Univariate and multivariate linear regression analysis revealed that LA length was significantly associated with LA longitudinal strain (Tables 5 and 6).

**Table 4. Clinical and echocardiographic characteristics in 53 patients with cardiovascular diseases.**

| Variable | Mean ± SD or Median (p25 to p75) |
| --- | --- |
| Age (year) | 70 (63 to 77) |
| Male/female | 38/15 |
| Body surface area (/m$^2$) | 1.66 (1.54 to 1.75) |
| Heart rate (bpm) | 68 ± 16 |
| Systolic blood pressure (mmHg) | 132 ± 28 |
| Diastolic blood pressure (mmHg) | 72 ± 12 |
| Clinical diagnosis | |
| Ischemic heart disease | 16 (31%) |
| Valvular heart disease | 9 (17%) |
| Secondary cardiomyopathy | 8 (15%) |
| Hypertensive heart disease | 7 (13%) |
| Dilated cardiomyopathy | 5 (10%) |
| Others | 7 (13%) |
| 2DE LV end-diastolic volume (mL) | 128 (100 to 192) |
| 2DE LV end-systolic volume (mL) | 71 (50 to 139) |
| 2DE LVEF (%) | 41 ± 13 |
| GLS (%) | 12.3 ± 4.7 |
| 2D LAVImax (mL/m$^2$) | 50.3 ± 18.5 |
| 2D LAVImin (mL/m$^2$) | 32.2 ± 17.5 |
| E wave velocity (cm/sec) | 68 ± 22 |
| A wave velocity (cm/sec) | 76 (61 to 98) |
| E/A | 0.83 (0.63 to 1.11) |
| s' average (cm/sec) | 6.7 ± 2.0 |
| e' average (cm/sec) | 7.1 ± 3.2 |
| E/e' average | 10.6 (7.5 to 12.7) |

2DE, 2-dimensional echocardiography; e', early diastolic peak mitral annular velocity; LVEF, left ventricular ejection fraction; p25, 25[th] percentile; p75, 75[th] percentile; s', peak systolic mitral annular velocity.

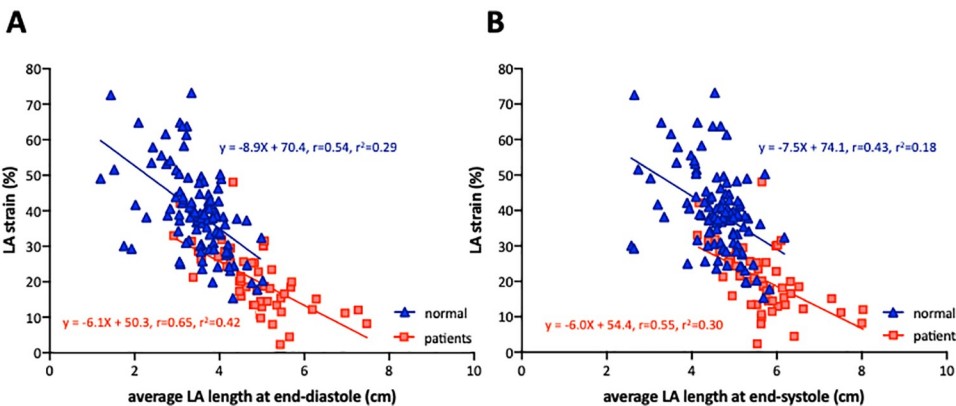

**Fig 7. Correlations between biplane LA strain and LA length in patients with cardiovascular disease (n = 53).** A: A linear correlation between average left atrial length at end-diastole and biplane left atrial longitudinal strain. B: A linear correlation between average left atrial length at end-systole and biplane left atrial longitudinal strain. Red rectangles represent patients. Data from normal subjects are superimposed (blue triangles).

**Table 5. Univariate and multivariate linear regression analysis for the association of anthropometric and echocardiography parameters with left atrial longitudinal strain assessed with 2D echocardiography in 53 patients having cardiovascular diseases.**

| | 4C LALS | | | | | | 2C LALS | | | | | | Biplane LALS | | | | | |
|---|---|---|---|---|---|---|---|---|---|---|---|---|---|---|---|---|---|---|
| | Univariable | | Multivariable 1* | | Multivariable 2† | | Univariable | | Multivariable 1* | | Multivariable 2† | | Univariable | | Multivariable 1* | | Multivariable 2† | |
| | t | p | t | p | t | p | t | p | t | p | t | p | t | p | t | p | t | p |
| Age | -2.90 | 0.005 | -3.66 | <0.001 | -3.62 | <0.001 | -2.38 | 0.020 | -2.44 | 0.018 | -2.34 | 0.023 | -2.89 | 0.005 | -3.56 | <0.001 | -3.47 | <0.001 |
| Male | 0.80 | 0.422 | | | | | 1.13 | 0.260 | | | | | 1.06 | 0.209 | | | | |
| BSA | 0.12 | 0.904 | | | | | 1.12 | 0.266 | | | | | 0.69 | 0.488 | | | | |
| HR | -2.24 | 0.028 | -3.50 | <0.001 | -2.66 | 0.010 | -1.34 | 0.185 | | | | | -1.92 | 0.059 | -2.90 | 0.005 | -2.39 | 0.020 |
| SBP | 1.16 | 0.248 | | | | | 0.74 | 0.461 | | | | | 1.02 | 0.308 | | | | |
| DBP | -0.33 | 0.735 | | | | | 0.13 | 0.890 | | | | | -0.09 | 0.923 | | | | |
| IQ good | -1.41 | 0.164 | | | | | -1.78 | 0.080 | | | | | -1.75 | 0.086 | | | | |
| IQ fair | -1.22 | 0.227 | | | | | -1.85 | 0.069 | | | | | -1.69 | 0.096 | | | | |
| IQ poor | -1.08 | 0.284 | | | | | -1.54 | 0.127 | | | | | -1.44 | 0.154 | | | | |
| FR | 0.60 | 0.549 | | | | | 1.70 | 0.095 | | | | | -1.15 | 0.249 | | | | |
| LA length max | -5.44 | <0.001 | -5.79 | <0.001 | | | -3.24 | 0.002 | -3.28 | 0.001 | | | -4.69 | <0.001 | -4.94 | <0.001 | | |
| LA length min | -6.59 | <0.001 | | | -6.42 | <0.001 | -4.31 | <0.001 | | | -4.25 | <0.001 | -6.04 | <0.001 | | | -5.92 | <0.001 |

Adjusted $R^2$ = 0.54, Adjusted $R^2$ = 0.58 Adjusted $R^2$ = 0.32, Adjusted $R^2$ = 0.31 Adjusted $R^2$ = 0.46, Adjusted $R^2$ = 0.53

* including LA length max.

† including LA length min.

## Observer variability

Intra-observer percent variability was as follows: 2DE LA longitudinal strain (8.3%), 3DE maximal LA volume (3.5%) and minimal LA volume (4.5%), 3DE LAEF (2.9%), and 3DE LA

**Table 6. Univariate and multivariate linear regression analysis for the association of left atrial longitudinal strain assessed with 3D echocardiography in patients having cardiovascular diseases.**

| | 3D LA longitudinal strain | | | | | |
|---|---|---|---|---|---|---|
| | Univariable | | Multivariable 1* | | Multivariable 2† | |
| | t-value | p-value | t-value | p-value | t-value | p-value |
| Age | -2.27 | 0.027 | -2.87 | 0.006 | -2.89 | 0.005 |
| Sex (male) | -0.83 | 0.410 | | | | |
| BSA | 0.20 | 0.834 | | | | |
| HR | -1.32 | 0.192 | | | | |
| SBP | 2.27 | 0.027 | 1.98 | 0.053 | 1.61 | 0.112 |
| DBP | -0.03 | 0.975 | | | | |
| IQ good | -0.33 | 0.741 | | | | |
| IQ fair | -0.61 | 0.543 | | | | |
| IQ poor | -0.28 | 0.773 | | | | |
| VR | 0.30 | 0.764 | | | | |
| Max. LA length | -3.05 | 0.003 | -2.57 | 0.013 | | |
| Min. LA length | -4.75 | <0.001 | | | -4.19 | 0.001 |

Adjusted $R^2$ = 0.40, p<0.001 Adjusted $R^2$ = 0.40, p<0.001

VR, volume rate. Other abbreviations were the same as Table 1.

* including LA length max.

† including LA length min.

**Table 7. Comparison of left atrial volumes and strain between LV-focused view and LA-focused view in 15 CMR cases.**

| | Maximal LAV (mL) | | | Minimal LAV (mL) | | | LA longitudinal strain (%) | | |
|---|---|---|---|---|---|---|---|---|---|
| | 4CV | 2CV | Biplane view | 4CV | 2CV | Biplane view | 4CV | 2CV | Biplane view |
| LV-focused view | 84 (63–136) | 78 (55–97) | 78 (68–116) | 54 (31–96) | 56 (29–86) | 51 (30–91) | 15.6 (5.1–23.7) | 16.8 (9.5–29.0) | 16.8 (9.7–22.1) |
| LA-focused view | 114 (89–154) | 86 (78–140) | 100 (83–147) | 75 (53–108) | 72 (53–110) | 72 (50–113) | 12.5 (5.5–16.4) | 11.2 (5.6–13.1) | 11.5 (6.9–14.2) |
| p | <0.0001 | 0.0026 | 0.0002 | 0.0001 | 0.0041 | 0.0002 | 0.0256 | 0.0003 | 0.0006 |
| r | 0.92 | 0.75 | 0.89 | 0.95 | 0.76 | 0.90 | 0.86 | 0.83 | 0.91 |

Data are expressed as median and (interquartile range).

4(2)CV, four-(two-)chamber view; LAV, left atrial volume.

longitudinal strain (10.5%). Corresponding ICCs were 0.88, 0.98, 0.97, 0.95, and 0.79, respectively. Corresponding levels of inter-observer percent variability were 9.5%, 7.7%, 7.9%, 4.6%, and 10.9%, respectively, and ICCs were 0.89, 0.93, 0.90, 0.90, and 0.73, respectively.

## CMR feature tracking

Table 7 compares maximal and minimal LA volumes and LA longitudinal strain measurements using views optimized for the left ventricle and those optimized for the left atrium. LA volumes were significantly larger when LA-focused views were used for the analysis. LA longitudinal strain was significantly lower when using LA-focused view than when using LV-focused view.

2DE-derived biplane maximal LAV (75 (53–91) mL) and minimal LAV (45 (22–53) mL) were significantly smaller than LA-focused CMR derived parameters (p = 0.002 and p = 0.0002, respectively). However, there was no significant difference between 2DE and LV-focused CMR-derived maximal LAV (p = 0.1239). 2DE derived biplane LALS (17.8 (15.0–27.0) %) was significantly higher than LV- and LA-focused CMR-derived LALS (p = 0.0101 and p = 0.0002, respectively).

## Acquisition of non-foreshortened LA images

LA long-axis diameter was significantly longer in non-foreshortened LA views compared with standard views (Fig 8). However, LA long-axis diameter in non-foreshortened LA view was still significantly shorter and LA strain was significantly larger than corresponding values measured using 3DE.

## Discussion

Major findings of this study are as follows: 1) Mean values of apical four-chamber and biplane LA longitudinal strain were 36.8% and 39.6%, respectively. 2) LA length at both end-diastole

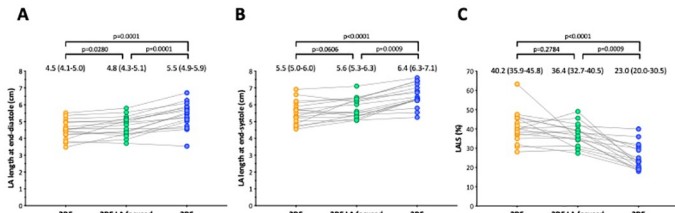

**Fig 8. Comparison of LA length and strain among standard 2D images, LA focused 2D images, and 3D echocardiography.** Values are expressed as medians and (interquartile ranges).

and end-systole was negatively correlated with biplane LA longitudinal strain. 3) Multivariate regression analysis revealed that both end-diastolic and end-systolic LA lengths still had a significant negative association after adjusting for anthropometric and echocardiographic image quality parameters. 4) 3DE LA longitudinal strain was significantly lower than 2DE LA longitudinal strain with a weak correlation. 5) LA length measured using 2DE was significantly shorter than that measured using 3DE. 6) Differences in LA length between 2DE and 3DE were an independent determinant of differences in LA longitudinal strain between the two methods. 7) The same trend was also observed in patients with cardiovascular disease. These results indicate that the left atrium often appears foreshortened in apical four-chamber and two-chamber views. This is significantly associated with overestimated 2DE LA longitudinal strain.

## Previous studies

LA longitudinal reservoir strain measurements are increasingly used in several clinical scenarios [3–7]. A recent meta-analysis revealed that the pooled reference value of LA reservoir strain was 39.4% in 2,542 healthy subjects [8]. However, the normal reservoir strain reported in each study ranged from 27.6% to 59.8%, with significant heterogeneity. In German and Japanese multicenter studies that enrolled >300 study subjects, the mean LA reservoir strain was 45.5 ±11.1% [14] and the median and IQR of LA reservoir strain were 42.5 (36.1–48.0) in a multicenter, European study [15]. Thus, normal LA reservoir strain was distributed over a wide range, contrasting with a reported reference range of LV global longitudinal strain and right ventricular free wall strain [16–18].

Few studies have addressed 3DE-determined LA reservoir longitudinal strain in healthy subjects. Mochizuki et al. performed both 2DE and 3DE LA strain analyses (Toshiba Medical Systems) in 75 healthy subjects, and reported that biplane LA longitudinal strain (35.8±7.7%) was significantly larger compared with 3DE LA longitudinal strain (28.1±7.4%, p<0.001) [19]. Recently, Nemes et al. determined reference values of 3DE LA longitudinal strain in healthy subjects (n = 222) using the same Toshiba software. They reported that mean values of 3DE LA reservoir longitudinal strain ranged from 24.0% in the younger age group to 28.6% in the middle-aged group [20].

## Current study

Mean values of apical four-chamber and biplane LA longitudinal strain were 36.8% and 39.6%, respectively, which are similar to the reported reference value of LA reservoir strain (39.4%) [8]. A paired comparison revealed that apical two-chamber LA longitudinal strain was significantly larger than apical four-chamber strain. However, LA length in the apical two-chamber view was significantly shorter than values measured in the apical four-chamber view at both end-diastole and end-systole. Linear regression analysis demonstrated that LA length was negatively correlated with apical four-chamber, two-chamber, and biplane LA longitudinal strain. These results support our hypothesis that 2DE LA length, especially minimal LA length negatively correlates 2DE LA strain measurements (shorter minimal LA length—higher 2DE LA strain) in healthy subjects.

3DE-determined LA longitudinal strain was significantly lower than LA longitudinal strain measured using 2DE with a weak correlation between the two parameters, which is in agreement with a previous study [19]. The same trend was also observed in patients with cardiovascular disease whose left atria were dilated in the majority of study patients.

Interestingly, there were no significant differences in LA volumes between 2DE and 3DE in healthy subjects. One potential explanation is that 2DE apical four- and two-chamber views

are frequently not orthogonal and both views cut near the major axis of the ellipsoid short axis view. This overestimates cross-sectional area of the LA short axis view, resulting in overestimation of LA volumes, even though LA length is underestimated [13].

LA length measured using 2DE was always shorter than values measured using 3DE. Thus, apical four-chamber and two-chamber views are foreshortened in the LA view, overestimating LA longitudinal strain.

Finally, we also confirmed our hypothesis using CMR SSFP images optimized for the left ventricle and for the left atrium. SSFP images aimed for the left atrium had significantly larger LA volumes and lower LA longitudinal strain than those optimized for the left ventricle. Our results verified that LV focused view is associated with overestimation of LA longitudinal strain. LA-focused CMR provided significantly larger LAVs and significantly lower LALS than the 2DE biplane method. From those results, LA images derived from CMR provides full delineation of the left atrium. Thus, we think CMR could be a reference method for LA volumetric analysis. CMR LALS was significantly lower than 2DE LALS; however, we cannot conclude that CMR should be a gold standard for LALS analysis because there is a difference in methodology between CMR feature tracking and 2D speckle tracking. Further study with a larger cohort incorporating CMR, 2DE, and prognostic analysis is required.

In this study, we did not exclude patients due to image quality because we wanted to assess the generalizability of our results to clinical situations. However, image quality could affect our results. In fact, 2DE average maximal LA length of normal subjects with poor or extremely poor image quality was significantly shorter than that of normal subjects with better image quality (4.30 ± 0.99 cm vs. 4.72 ± 0.53 cm, p = 0.011). Therefore, we have performed additional analyses, and confirmed that the results of only patients with good or fair image quality were nearly identical with the main results (data are not shown).

Recently, Unlu et al. reported that foreshortened LV results in large LVGLS, mainly because of overestimation of apical longitudinal strain [21]. We think a similar phenomenon was observed in LA analysis; therefore, overestimation in the roof portion of LA may be responsible for the overestimation of global LALS. Further study focused on reginal analysis is required.

## Clinical implications

To eliminate the foreshortened LA view, separate acquisition of ideal apical views that allows full visualization of the LA cavity is essential. Some studies have stressed the importance of acquiring an atrial focused view [10, 11]. However, this is not always easy because there are no anatomical landmarks to guide the practitioner. 2DE does not provide a third dimension; thus, there are no clues to ensure that a 2DE view aimed at the left atrium actually evaluated the maximal LA long-axis. Angulation between the LV long axis and the LA long axis makes a suitable apical recoding point for the left atrium more caudally oriented. This kind of image acquisition is not possible in every subject (Fig 9). Results from our prospective acquisition of non-foreshortened LA view support this difficulty. Since the left atrium is located far from the transducer using an apical approach, 3DE datasets encompass the entire left atrium, allowing full delineation of the LA long-axis in every subject. Thus, 3DE assessment of LA strain can overcome the problem of foreshortening. If 3DE is not available, we recommend using a 4CV single plane assessment because of its reduced foreshortening and LALS overestimation.

## Study limitations

There were several limitations that should be acknowledged. First, there is no 'gold standard' for LA longitudinal strain measurements; thus, we are unable to comment on which method

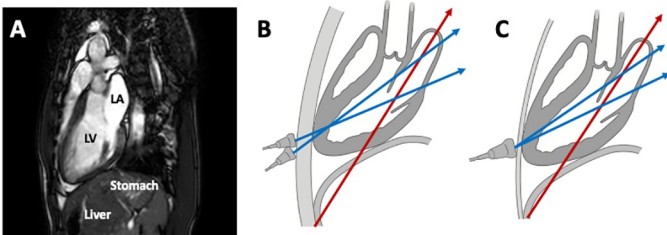

**Fig 9. Difficulty in obtaining an optimized LA view.** Panel A shows a cardiac magnetic resonance image of the heart and surrounding structures. Panel B shows a subject in whom there is some distance between the chest wall and the heart apex. This is typical of large-bodied westerners. In such cases, there is space on the chest wall to record both the left ventricle and the left atrium (blue arrows). However, the LA long axis runs more caudally (red arrow); thus, it is quite difficult to obtain an optimal LA view. Panel C depicts a subject whose apex is located behind the chest wall. This is typical of slender Japanese subjects. Since the scanning area is limited, only angulation of the transducer enables a different cut-plane of the left atrium. However, it is difficult to make significant changes in long axis delineation of the LA wall. Delineation changes cause subtle changes in both the LA long axis diameter and LA strain.

is more accurate. However, the main aim of this study was to demonstrate that LA longitudinal strain using 2DE speckle tracking analysis is overestimated in healthy subjects. Second, we used endocardial LA strain for comparisons, whereas the majority of studies have used mid-myocardial strain. However, when we used mid-myocardial strain, the trend was the same (data not shown) because the transmural variation of strain across the LA wall is calculated by assuming a linear distribution [22]. Third, the correlations between 2DE biplane LALS and 2DE LA length at end-diastole or end-systole were relatively small. We cannot determine whether the reason was the small sample size or the weakness of the relationships. However, according to multivariate analyses, LA lengths were still significantly associated with LALS even after adjusting for a number of covariates. We think this result supports the importance of LA length on LALS. The same study limitation applies to the association of the difference in LA strain between 2DE and 3DE and the difference in LA lengths between 2DE and 3DE. Fourth, in 2DE analysis of 105 normal subjects, there was a possibility that all LA images were foreshortened because this is a retrospective cohort and recorded standard apical views were aimed at both LV and LA. This is why we prospectively collected 20 additional healthy subjects to acquire LA-focused images, and we confirmed that our hypothesis was correct. Fifth, the correlations of LAEF and LALS between 2DE and 3DE were poor. This might be caused by the anatomical complexity of LA and the difference in the degree of foreshortening in 2DE, case by case. Sixth, supraventricular arrhythmia such as atrial fibrillation also could affect the degree of LALS overestimation. However, there were only 3 patients with atrial fibrillation in our cohort, too few to statistically analyze the effect of arrhythmia. In the future, further study is required including a large cohort with arrhythmias. Seventh, we have used 3DE and CMR for reference; however, those also have limitations for spatial and temporal resolution. Finally, results were obtained using software from a single ultrasound vendor and it is premature to say whether our results can be generalized to software from other vendors.

## Conclusions

Reference values of LA longitudinal reservoir strain determined with 2DE have the risk of being overestimated due to foreshortening of the LA cavity in standard apical views. Since acquisition of non-foreshortened LA views is not easy, 3DE assessments of LA longitudinal strain may overcome this problem.

## Supporting information

**S1 Data.**
(ZIP)

**S2 Data.**
(XLSX)

**S3 Data.**
(XLSX)

**S4 Data.**
(XLSX)

**S5 Data.**
(XLSX)

**S6 Data.**
(XLSX)

## Author Contributions

**Conceptualization:** Masaaki Takeuchi.

**Data curation:** Yosuke Nabeshima, Tetsuji Kitano, Masaaki Takeuchi.

**Formal analysis:** Yosuke Nabeshima, Masaaki Takeuchi.

**Methodology:** Masaaki Takeuchi.

**Project administration:** Masaaki Takeuchi.

**Supervision:** Masaaki Takeuchi.

**Validation:** Masaaki Takeuchi.

**Visualization:** Yosuke Nabeshima.

**Writing – original draft:** Yosuke Nabeshima.

**Writing – review & editing:** Masaaki Takeuchi.

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
