## [Decision Letter · Decision Letter 0]

23 Dec 2020

PONE-D-20-30699

Reliability of left atrial strain reference values: A 3D echocardiographic study

PLOS ONE

Dear Dr. Nabeshima,

Thank you for submitting your manuscript to PLOS ONE. After careful consideration, we feel that it has merit but does not fully meet PLOS ONE’s publication criteria as it currently stands. Therefore, we invite you to submit a revised version of the manuscript that addresses the points raised during the review process.

We look forward to receiving your revised manuscript.

Kind regards,

Kumaradevan Punithakumar

Academic Editor

PLOS ONE

Journal Requirements:

2. Please amend your list of authors on the manuscript to ensure that each author is linked to an affiliation. Authors’ affiliations should reflect the institution where the work was done (if authors moved subsequently, you can also list the new affiliation stating “current affiliation:….” as necessary).

Reviewers' comments:

Reviewer's Responses to Questions

**Comments to the Author**

1. Is the manuscript technically sound, and do the data support the conclusions?

Reviewer #1: Partly

Reviewer #2: Yes

2. Has the statistical analysis been performed appropriately and rigorously? 

Reviewer #1: N/A

Reviewer #2: Yes

3. Have the authors made all data underlying the findings in their manuscript fully available?

Reviewer #1: Yes

Reviewer #2: Yes

4. Is the manuscript presented in an intelligible fashion and written in standard English?

Reviewer #1: Yes

Reviewer #2: Yes

5. Review Comments to the Author

Reviewer #1: In this manuscript, the author measured the LA length and LALS using 2DE and 3DE, respectively, in 105 healthy subjects and 53 patients with cardiovascular disease. The results showed that LA length had a significant negative relationship with biplane LALS, 3D LALS were significantly lower than 2D LALS, while LA length measured by 3D was longer than 2D. Based on the results, they draw the conclusion that in 2DE, the LA cavity consistent. It is a great study, but some issues need to be addressed.

1. The way that 2DE and 3DE calculate the LALS is different. The different LALS measured by 2DE and 3DE might not be caused by LA length foreshortening.

2. this study aimed to evaluate if 2DE speckle tracking analysis overestimates the reference value of LALS, but the first aim “determine the impact of LA length on 2DE LA longitudinal strain values” of the study were to determine the impact of LA length on 2DE LALS values. The LA length and LALS measured using the AP2 view is significantly short and higher than using the AP4 view, respectively. That could support the hypothesis. However, the negative correlation (figure 4, table 2, and table 3) between LA length and LALS in the cohort could not prove that foreshortening of LA length would lead to LALS overestimate, which could only indicate that the shorter the LA length for a subject, the higher the LALS based on a different subject with distinct LA length in the study cohort.

And the r for the biplane LALS and LA length at either end-diastolic or systolic was relatively small, and r square could be given as well.

3. The image quality was graded from good to fair, poor, and extremely poor. In 2DE imaging, there were 24 (23%) subjects had poor image quality, while 11 (10%) subjects had poor image quality in 3DE imaging. Because the poor imaging would affect the accuracy of the LALS measurement, why those patients with extremely poor imaging and/or poor imaging were not excluded from the study? The inaccurate measurement of LA length and LALS would lead to the negative correlation between LA length and LALS unsolid, as well we the comparison between 2DE LA parameters and 3DE LA parameter. And how to define the feasibility (the imaging could be trace or could be clearly/accurate trace)? Even if the LALS could be measured from the extremely poor LA imaging (100% feasibility) with reluctance, the accuracy, which is more important than feasibility, could not be as high as 100%.

What is more, the imaging quality should not be used as a variable in linear regression analysis for LALS because, for the same reason mention before, the extremely poor could either overestimate or underestimate the LA length and LALS and not in a consistent pattern.

Same problems in patients with cardiovascular disease.

4. In the method, 20 health patients were enrolled to get the non-foreshortened LA imaging. Please add descriptions about “non-foreshortened and standard LA view”. Did it mean that in “2DE analysis”, the LA measurements were all foreshortened because of using the standard view?

5. Why tables 2, 3 and 5 had two multivariable analysis columns? What was the multivariable 1 and 2 in table 3 represent for? And the adjusted R square was small.

5. How consistent were the 2DE LA parameters and the CMR LA parameters? Was CMR the golden standard or a more solid method for LA measurement?

Minor:

6. The absolute difference between 2DE and 3DE LA strain correlated with differences in LA lengths between 2DE and 3DE at both end-diastole and end-systole, but the r was too small, which indicate the week or not solid relationship even the P-value was lower than 0.01.

7. Please add more discussion about the possible reasons why 2DE could overestimate LALS and underestimate the LA length. And the poor correlation between LAEF and LALS using 2DE or 3DE.

8. Evidence of no collinearity of every variable could be provided before multivariate linear regression analyses.

9. The avoidance of the interference by pulmonary vein when measure the LA parameters should be mentioned in the "method".

10. Were there any subjects in the whole cohort that have an arrhythmia, which might affect the LALS as well?

Reviewer #2: 1. Study populations: Please clarify (in abstract and other sections) that there were 3 groups of patients studied: 1. Normal 105 (retrospectively), 2. 53 with CVD (retrospectively), and 3. 15 with CMR (prospectively, IRB?);

2. It would be reasonable to compare 2D vs 3D, 2D echo vs 2D CMR for LA size and strain, using 3D and CMR as "references". But 3D and CMR LA size and strain may also be limited for the spatial and temporal resolution therefore not "gold standard.

3. There appeared to be "associations" between LA size and strain, but they may not be cause and effect relationship in the text of the manuscript.

4. Without 3D, would it be more appropriate to use 4CV, 2CV or biplane to measure LA size and strain? Why and why not?

6. PLOS authors have the option to publish the peer review history of their article (what does this mean?). If published, this will include your full peer review and any attached files.

Reviewer #1: No

Reviewer #2: No

---

## [Author Response · Author response to Decision Letter 0]

4 Mar 2021

We thank the reviewers for their very thoughtful and constructive comments and suggestions. 

Please see the separated file named "Respond to Reviewers", because text layout collapses.

Responses to the Reviewer #1:

In this manuscript, the author measured the LA length and LALS using 2DE and 3DE, respectively, in 105 healthy subjects and 53 patients with cardiovascular disease. The results showed that LA length had a significant negative relationship with biplane LALS, 3D LALS were significantly lower than 2D LALS, while LA length measured by 3D was longer than 2D. Based on the results, they draw the conclusion that in 2DE, the LA cavity consistent. It is a great study, but some issues need to be addressed.

1. The way that 2DE and 3DE calculate the LALS is different. The different LALS measured by 2DE and 3DE might not be caused by LA length foreshortening.

We appreciate this comment. We asked a technical staff member at GE software about this issue, and he responded that the 3D quantification method (LAQ) uses endocardial border detection to track the motion of the LA wall. Strain can then be computed from the change in length of the endocardial border = (L(end-systole) - L(end-diastole))/L(end-diastole) x 100%. 2D strain uses speckle tracking of the myocardium to track the motion of the LA wall. In this case, strain is then computed using the same formula as for LAQ, from the length of the endocardial line of the ROI. Thus, both types of software use the same formula to calculate LA strain. Of course, while there is a geometrical difference between 2DE and 3DE, the basic methods are the same. We have clearly explained this point in the Methods section, as shown below.

Page 7-8, 3DE analysis

In this study, both 2D and 3D strain analysis software employed the same formula to compute the change in length (L) of the endocardial border = (L(end-systole) – L(end-diastole)) / L(end-diastole) x 100%.

2. this study aimed to evaluate if 2DE speckle tracking analysis overestimates the reference value of LALS, but the first aim “determine the impact of LA length on 2DE LA longitudinal strain values” of the study were to determine the impact of LA length on 2DE LALS values. The LA length and LALS measured using the AP2 view is significantly short and higher than using the AP4 view, respectively. That could support the hypothesis. However, the negative correlation (figure 4, table 2, and table 3) between LA length and LALS in the cohort could not prove that foreshortening of LA length would lead to LALS overestimate, which could only indicate that the shorter the LA length for a subject, the higher the LALS based on a different subject with distinct LA length in the study cohort.

We are grateful for this constructive comment. As the reviewer commented, Figure 4, Table 2, and Table 3 indicated that subjects the shorter LA length were associated with higher LALS after adjusting covariates that affect LA size and function. Therefore, we have to indicate whether foreshortened LA is associated with higher LALS and LALS overestimation to prove our hypothesis. The association between foreshortened LA and possible overestimation of LA strain was already described in the third paragraph of the section entitled “Comparisons of 2DE and 3DE analyses.” Thus, we performed two additional multivariate regression analyses to reveal the association between LA foreshortening and higher LALS.

Li et al. reported that the lower limit of longitudinal LA length at end-systole of an Asian population was 3.8 cm in 2ch and 4.5 cm in 4ch [1]. If we define that averaged 2DE LA length lower than ((3.8+4.5)/2=) 4.2 cm as too short, meaning possible foreshortening of LA, 19 of 94 normal subjects showed underestimated LA.” Subjects with underestimated LA showed significantly higher LALS (47±13%) than others (37±11%) (p=0.003). According to multivariate linear regression analysis, underestimated LA was significantly associated with higher LALS, even after adjusting for age, sex, BSA, SBP, heart rate, image quality, and 3DE maximal LAV (p=0.047, t=2.01). We think this result indicates foreshortened LA is associated higher LALS, independent of the real LA size and other characteristics of the subject.

We also performed additional multivariate linear regression analyses. The differences in LA length at both end-diastole (p<0.001) and end-systole (p=0.002) had a significant association with 2DE LA strain after adjusting for age, sex, systolic blood pressure, volume rate, 3DE image quality, and minimal or maximal 3DE LAV. This indicates that foreshortened LA itself is significantly associated with 2DE LALS measurements, independent of LA size and other covariates.

These results indicate that LA foreshortening is at least one of the factors associated with higher 2DE LALS. Thus, we added the text below to the Results section.

Page 15, Comparisons of 2DE and 3DE analyses

The difference in LA lengths between 2DE and 3DE at both end-diastole (r=0.21, p<0.001) and end-systole (r=0.30, p<0.001) was modestly correlated with 2DE LA strain. Multivariate linear regression analysis revealed that differences in LA length at both end-diastole (p<0.001) and end-systole (p=0.002) had a significant association with 2DE LA strain after adjusting for age, sex, systolic blood pressure, volume rate, 3DE image quality, and minimal or maximal 3DE LAV.

1. Li W, Wan K, Han Y, Liu H, Cheng W, Sun J, et al. Reference value of left and right atrial size and phasic function by SSFP CMR at 3.0 T in healthy Chinese adults. Sci Rep. 2017;7(1):3196. Epub 2017/06/11. doi: 10.1038/s41598-017-03377-6. PubMed PMID: 28600567; PubMed Central PMCID: PMCPMC5466635.

And the r for the biplane LALS and LA length at either end-diastolic or systolic was relatively small, and r square could be given as well.

We are grateful for this important comment. We modified the figure and added a sentence to the limitation section, as shown below.

Page 29, Study limitations

Third, the correlations between 2DE biplane LALS and 2DE LA length at end-diastole or end-systole were relatively small. We cannot determine whether the reason was the small sample size or the weakness of the relationships. However, according to multivariate analyses, LA lengths were significantly associated with LALS even after adjusting for a number of covariates. We think this result supports the importance of LA length on LALS.

3. The image quality was graded from good to fair, poor, and extremely poor. In 2DE imaging, there were 24 (23%) subjects had poor image quality, while 11 (10%) subjects had poor image quality in 3DE imaging. Because the poor imaging would affect the accuracy of the LALS measurement, why those patients with extremely poor imaging and/or poor imaging were not excluded from the study? The inaccurate measurement of LA length and LALS would lead to the negative correlation between LA length and LALS unsolid, as well we the comparison between 2DE LA parameters and 3DE LA parameter. And how to define the feasibility (the imaging could be trace or could be clearly/accurate trace)? Even if the LALS could be measured from the extremely poor LA imaging (100% feasibility) with reluctance, the accuracy, which is more important than feasibility, could not be as high as 100%.

What is more, the imaging quality should not be used as a variable in linear regression analysis for LALS because, for the same reason mention before, the extremely poor could either overestimate or underestimate the LA length and LALS and not in a consistent pattern.

Same problems in patients with cardiovascular disease.

We appreciate these critical comments. Because we considered the generalizability of our results for clinical situation, we did not exclude subjects due to poor image quality. However, as the reviewer noted, poor image quality could affect results. In fact, 2DE average maximal LA length of normal subjects with poor or extremely poor image quality was significantly shorter than in normal subjects with better image quality (4.30 ± 0.99 cm vs. 4.72 ± 0.53 cm, p=0.011).

Therefore, we performed subgroup analysis that included normal subjects with only good or fair image quality for both 2DE and 3DE (N=65). There were significant inverse correlations between biplane LA longitudinal strain and average LA length at end-diastole (r=0.58, p<0.001) and between biplane LA longitudinal strain and average LA length at end-systole (r=0.43, p<0.001). LA longitudinal strain using 3DE (24.3 ± 8.0%) was significantly lower than that derived using 2DE (40.6 ± 10.9%, p<0.001) and the two values were moderately correlated (r=0.47). 2DE-determined LA length was significantly shorter than 3DE-determined LA length at both end-diastole (3.57 ± 0.62 cm vs. 4.82 ± 0.57 cm, p<0.001) and end-systole (4.71 ± 0.55 cm vs. 5.81 ± 0.54 cm, p<0.001). The results of multivariate regression analyses were as shown below (Table 1’). Both minimal and maximal LA lengths were associated with biplane LALS, even after adjusting for age, sex, BSA, HR, SBP, and 3D LAV. 

We also performed subgroup analysis that included patients with cardiovascular diseases having only good or fair image quality for both 2DE and 3DE (N=36). There were significant inverse correlations between biplane LA longitudinal strain and average LA length at end-diastole (r=0.58, p<0.001) and between biplane LA longitudinal strain and average LA length at end-systole (r=0.46, p<0.001). LA longitudinal strain using 3DE (12.8 ± 7.4%) was significantly lower than values derived using 2DE (19.6 ± 9.2%, p<0.001) and the two values were correlated (r=0.81). 2DE-determined LA length was significantly shorter than 3DE-determined LA length at both end-diastole (4.99 ± 0.96 cm vs. 6.02 ± 0.93 cm, p<0.001) and end-systole (5.83 ± 0.80 cm vs. 6.63 ± 0.78 cm, p<0.001). The results of multivariate regression analyses were as shown below (Table 2’). Both minimal and maximal LA lengths were associated with biplane LALS, even after adjusting for age, HR, and 3D LAV.

Table 1’: Multivariate linear regression analysis for the association of left atrial longitudinal strain assessed by 2D echocardiography in 65 normal subjects with good or fair image quality

 2D biplane LA longitudinal strain (N=65)

 Multivariable 1* Multivariable 2†

 t-value p-value t-value p-value

Age -3.614 <0.0001 -2.254 0.0281

Sex (male) 

BSA 

HR 

SBP 

3DE LAVmax 2.725 0.0086 

3DE LAVmin 

2DE Max. LA length -3.185 0.0024 

2DE Min. LA length -3.621 0.0006

 Adjusted R2 = 0.321, p<0.0001 Adjusted R2 = 0.330, p<0.0001

* including LA length max 

† including LA length min 

 

Table 2’: Multivariate linear regression analysis for the association of left atrial longitudinal strain assessed by 2D echocardiography in 36 patients with good or fair image quality

 2D biplane LA longitudinal strain (N=36)

 Multivariable 1* Multivariable 2†

 t-value p-value t-value p-value

Age -3.642 0.0007 -3.600 0.0008

HR -2.782 0.0079 -2.174 0.0350

3DE LAVmax 

3DE LAVmin -2.149 0.0371

2DE Max. LA length -2.259 0.0288 

2DE Min. LA length -2.644 0.0112

 Adjusted R2 = 0.470, p<0.0001 Adjusted R2 = 0.574, p<0.0001

* including LA length max 

† including LA length min 

Those results are almost identical with the results for the whole cohort. Thus, we added an explanation about image quality to the Discussion section.

Page 27, Current study

In this study, we did not exclude patients due to image quality because we wanted to assess the generalizability of our results to clinical situations. However, image quality could affect our results. In fact, 2DE average maximal LA length of normal subjects with poor or extremely poor image quality was significantly shorter than that of normal subjects with better image quality (4.30 ± 0.99 cm vs. 4.72 ± 0.53 cm, p=0.011). Therefore, we have performed additional analyses, and confirmed that the results of only patients with good or fair image quality were nearly identical with the main results (data are not shown).

We also added an explanation about feasibility to the Method section.

Page 6, 2DE analysis

We determined the feasibility of 2DE speckle tracking software according to the quality of tracking after manual endocardial tracing.

Page 8, 3DE analysis

We determined the feasibility of 3DE speckle tracking software according to the possibility of adequate tracking after manual correction of automatically detected endocardial border as required.

4. In the method, 20 health patients were enrolled to get the non-foreshortened LA imaging. Please add descriptions about “non-foreshortened and standard LA view”. Did it mean that in “2DE analysis”, the LA measurements were all foreshortened because of using the standard view?

We appreciate this important comment. Considering 2DE analyses, because of the retrospective nature of this study, we cannot ensure that the acquired standard apical images were the best for LA. Most of apical images were better for both LA and LV, as in Figures 1A and 1B. So there was a possibility that all LA images were foreshortened. This is why we prospectively collected an additional 20 healthy subjects, and we confirmed that our hypothesis was supported. We added the text below to the limitation section.

Page 28, Study limitations

Fourth, in 2DE analysis of 105 normal subjects, there was a possibility that all LA images were foreshortened because this is a retrospective cohort and recorded standard apical views were aimed at both LV and LA. This is why we prospectively collected 20 additional healthy subjects to acquire LA-focused images, and we confirmed that our hypothesis was correct.

5. Why tables 2, 3 and 5 had two multivariable analysis columns? What was the multivariable 1 and 2 in table 3 represent for? 

We are sorry about the confusion. We performed two types of multivariable analyses for LA max length and LA min length, respectively. Because there was collinearity between LA max length and LA min length, we had to perform multivariable analyses separately. We modified the explanations and tables to avoid confusion. 

Page 13, 2DE analysis; Page 16, 3DE analysis; Page 18, Sensitivity analysis

Because there was collinearity between the minimal and maximal LA lengths, we performed two types of multivariate linear regression analysis, including minimal or maximal LA length.

And the adjusted R square was small.

We performed multivariable analyses for determining whether LA length was associated with LALS even after adjusting for covariates. The small R2 value indicated that the overall multivariate model is not good for predicting LALS; however, it does not eliminate the significance of LA length. If appropriate, we will add an explanation about this point to the limitation section.

5. How consistent were the 2DE LA parameters and the CMR LA parameters? Was CMR the golden standard or a more solid method for LA measurement?

We agree this is a very important point and we analyzed the differences of measurements between LA focused CMR, LV focused CMR and 2DE using Friedman test and Dunn’s test. 2DE-derived biplane maximal LAV (75 (53-91) mL) and minimal LAV (45 (22-53) mL) were significantly smaller than LA-focused CMR derived parameters (p=0.002 and p=0.0002, respectively). However, there was no significant difference between 2DE and LV-focused CMR-derived maximal LAV (p=0.1239). 2DE-derived biplane LALS (17.8 (15.0-27.0) %) was significantly higher than LV- and LA-focused CMR-derived LALS (p=0.0101 and p=0.0002, respectively). From those results, LA images derived from CMR are more similar to whole images of LA. CMR LALS was significantly lower than 2DE LALS; however, we cannot conclude that CMR LALS should be the gold standard because there was a significant difference in methodology and CMR also has limitations. We added an explanation about these points to the Methods, the Results and the Discussion sections.

Page 9, Validation study using cardiac magnetic resonance

We also compared the LAVs and LALS between the CMR biplane method and the 2DE biplane method.

Page 22, CMR feature tracking

2DE-derived biplane maximal LAV (75 (53-91) mL) and minimal LAV (45 (22-53) mL) were significantly smaller than LA-focused CMR derived parameters (p=0.002 and p=0.0002, respectively). However, there was no significant difference between 2DE and LV-focused CMR-derived maximal LAV (p=0.1239). 2DE derived biplane LALS (17.8 (15.0-27.0) %) was significantly higher than LV- and LA-focused CMR-derived LALS (p=0.0101 and p=0.0002, respectively).

Page 26-27, Current study

LA-focused CMR provided significantly larger LAVs and significantly lower LALS than the 2DE biplane method. From those results, LA images derived from CMR seemed to be more similar to whole images of LA. Thus, we think CMR could be a reference method for LA volumetric analysis. CMR LALS was significantly lower than 2DE LALS; however, we cannot conclude that CMR should be a gold standard for LALS analysis because there is a difference in methodology between CMR feature tracking and 2D speckle tracking. Further study with a larger cohort incorporating CMR, 2DE, and prognostic analysis is required.

Minor:

6. The absolute difference between 2DE and 3DE LA strain correlated with differences in LA lengths between 2DE and 3DE at both end-diastole and end-systole, but the r was too small, which indicate the week or not solid relationship even the P-value was lower than 0.01.

As the reviewer noted, the relationship may be weak; however, multivariate linear regression analysis revealed that differences in LA length at both end-diastole and end-systole had a significant association with differences in LA longitudinal strain after adjusting for covariates. So we believe there is an association between differences in LA length and differences in LALS. This point is similar to the answer for Major-2, and we added the following sentence to the limitation section. We also modified the Results.

Page 15, Comparisons of 2DE and 3DE analyses

The absolute difference between 2DE and 3DE LA strain was 15.7±12.0%, which was modestly correlated with differences in LA lengths between 2DE and 3DE at both end-diastole (r=0.29, p=0.004) and end-systole (r=0.31, p<0.003).

Page 29, Study limitations

The same study limitation applies to the association of the difference in LA strain between 2DE and 3DE and the difference in LA lengths between 2DE and 3DE.

7. Please add more discussion about the possible reasons why 2DE could overestimate LALS and underestimate the LA length. And the poor correlation between LAEF and LALS using 2DE or 3DE.

As Unlu et al. reported, foreshortened LV results in large LVGLS. Especially, apical and mid-ventricular segments were more affected by foreshortening [2]. We think the same phenomenon was observed in LA analysis. A possible reason why 2DE underestimates the LA length is indicated in Figure 9. Especially in patients with thin chests, the long axis of LA is more vertical. In such a case, the ultrasonic beam must pass through the diagram via abdomen (red arrow) to obtain a non-foreshortened LA image. This is technically difficult and the resulting image quality should be worse. However, if we acquire the LA image using a thoracic view (blue arrow), foreshortening is unavoidable. As in LV, foreshortening causes overestimation of LALS especially in the LA roof portion. Thus, we have added the text below to the Discussion section.

Page 27, Current study

Recently, Unlu et al. reported that foreshortened LV results in large LVGLS, mainly because of overestimation of apical longitudinal strain [21]. We think a similar phenomenon was observed in LA analysis; therefore, overestimation in the roof portion of LA may be responsible for the overestimation of global LALS. Further study focused on reginal analysis is required.

The poor correlations of LAEF and LALS between 2DE and 3DE may be caused by the anatomical complexity of the LA. Because the LA is not an ellipsoidal sphere, there might be a methodological limitation to the biplane Simpson’s method to calculate 2DE-derived LAVs. The reason why correlation between 2DE and 3DE LALS was poor may be that there was a difference in the degree of foreshortening in 2DE, case by case. We have added the explanation below to the limitation section.

Page 29, Study limitations

Fifth, the correlations of LAEF and LALS between 2DE and 3DE were poor. This might be caused by the anatomical complexity of LA and the difference in the degree of foreshortening in 2DE, case by case.

2. Unlu S, Duchenne J, Mirea O, Pagourelias ED, Bezy S, Cvijic M, et al. Impact of apical foreshortening on deformation measurements: a report from the EACVI-ASE Strain Standardization Task Force. Eur Heart J Cardiovasc Imaging. 2020;21(3):337-43. Epub 2019/07/31. doi: 10.1093/ehjci/jez189. PubMed PMID: 31361311.

8. Evidence of no collinearity of every variable could be provided before multivariate linear regression analyses.

We confirmed that the factors used for multivariate analysis showed r values lower than 0.70. LA max and min lengths showed collinearity and we did not include those factors at the same time. We clearly indicated the collinearity between minimal and maximal LA length as given above (Major-5).

Page 13, 2DE analysis; Page 16, 3DE analysis; Page 18, Sensitivity analysis

Because there was collinearity between the minimal and maximal LA lengths, we performed two types of multivariate linear regression analysis, including minimal or maximal LA length.

9. The avoidance of the interference by pulmonary vein when measure the LA parameters should be mentioned in the "method".

This is an important point. We added sentences to the Methods section.

Page 6, 2DE analysis

Pulmonary veins and left atrial appendages were carefully excluded from the endocardial border.

Page 7, 3DE analysis

As addition to 2DE analysis, pulmonary veins and left atrial appendages were carefully excluded from the endocardial border.

10. Were there any subjects in the whole cohort that have an arrhythmia, which might affect the LALS as well?

Only three patients had atrial fibrillation and all other patients had sinus rhythm. In patients with Afib, LALS was lower and LA length was longer compared to patients with sinus rhythm (Table 3’). Because only three patients had Afib, it was difficult to analyze the statistical significance of the effect of arrhythmia on the Results. Thus, we added a sentence explaining this to the limitation section.

Table 3’: LA measurements of patients with sinus rhythm and atrial fibrillation

 Sinus (N=50) Afib (N=3)

IQ good/fair/poor 11/23/16 1/1/1

2D Biplane LA length max (cm) 5.6 (5.0-6.0) 8.0 (7.3-8.0)

2D Biplane LA length min (cm) 4.8 (4.3-5.2) 7.3 (6.3-7.5)

3D LA length max (cm) 6.4 (5.8-6.9) 8.1 (7.7-8.5)

3D LA length min (cm) 5.7 (5.1-6.3) 7.6 (7.1-8.2)

3D LAV max (mL) 64 (54-73) 135 (126-143)

3D LAV min (mL) 38 (28-48) 108 (104-111)

LA length max difference (3D-2D) (cm) 0.8 (0.4-1.3) 0.45 (0.43-0.46)

LA length min difference (3D-2D) (cm) 1.0 (0.5-1.5) 0.8 (0.7-0.8)

2D Biplane LALS (%) 20.1 (14.2-26.6) 12.1 (8.2-15.2)

3D LALS (%) 13.0 (9.3-18.0) 3.5 (2.0-5.0)

LALS difference (2D-3D) (%) 7.6 (1.6-10.6) 8.2 (6.2-10.2)

Page 29, Study limitation

Sixth, supraventricular arrhythmia such as atrial fibrillation also could affect the degree of LALS overestimation. However, there were only 3 patients with atrial fibrillation in our cohort, too few to statistically analyze the effect of arrhythmia. In the future, further study is required including a large cohort with arrhythmias.

Responses to the Reviewer #2

1. Study populations: Please clarify (in abstract and other sections) that there were 3 groups of patients studied: 1. Normal 105 (retrospectively), 2. 53 with CVD (retrospectively), and 3. 15 with CMR (prospectively, IRB?);

We appreciate this critical comment. We apologize that we forgot to include the details of prospective cohorts. We have now added text to the Abstract and the Methods.

Page 2, Abstract

In this study, 4 types of cohorts were included: 1. 105 normal subjects (retrospectively), 2. 53 patients with cardiovascular diseases (retrospectively), 3. 15 patients who received cardiac magnetic resonance (prospectively), and 4. 20 normal subjects (prospectively).

Page 5, Study population

We retrospectively selected 105 healthy adult volunteers (≥20 years old) who had undergone both 2DE and 3DE examinations using ultrasound equipment from a single manufacturer (GE Healthcare, Horten, Norway) from our 3DE database to determine reference values for 2DE and 3DE LA longitudinal strain. To determine whether results are consistently observed in patients, we also selected from the database, 53 adult patients with cardiovascular disease who underwent both clinical 2DE and 3DE examinations. For the sensitivity analyses, we prospectively collected 15 adult patients who were clinically indicated for cardiac magnetic resonance and 20 healthy adult volunteers. The Ethics committee approved the study protocol. The need to obtain informed consent was waived for the retrospective cohorts. Informed consent was obtained from prospective cohorts.

2. It would be reasonable to compare 2D vs 3D, 2D echo vs 2D CMR for LA size and strain, using 3D and CMR as "references". But 3D and CMR LA size and strain may also be limited for the spatial and temporal resolution therefore not "gold standard.

We completely agree with the reviewer. We added sentences about 3D and CMR to the limitation section.

Page 29, Study limitations

Seventh, we have used 3DE and CMR for reference; however, those also have limitations for spatial and temporal resolution.

3. There appeared to be "associations" between LA size and strain, but they may not be cause and effect relationship in the text of the manuscript.

We appreciate this important comment. As the reviewer suggested, we revealed only associations between foreshortened LA and LALS overestimation. We deleted and rephrased the text which indicated a causal relationship between LA size and function.

Page 9, Validation study using cardiac magnetic resonance

To validate the relationship between the LA foreshortened view and LA strain analysis, we prospectively acquired apical four-chamber and two-chamber, steady-state free precession (SSFP), dynamic gradient-echo cine loops optimized for the left ventricle, and those optimized for the left atrium in 15 patients who had clinically indicated cardiac magnetic resonance (CMR) examinations.

Page 24, Discussion

This is significantly associated with overestimated 2DE LA longitudinal strain.

Page 26, Current study

These results support our hypothesis that 2DE LA length, especially minimal LA length negatively correlates 2DE LA strain measurements (shorter minimal LA length – higher 2DE LA strain) in healthy subjects.

4. Without 3D, would it be more appropriate to use 4CV, 2CV or biplane to measure LA size and strain? Why and why not?

We think that the 4CV single plane method is more appropriate, because 2CV was more frequently foreshortened, and the degree of LALS overestimation was larger both in healthy and diseased subjects. We have added a sentence to the clinical implication section.

Page 28, Clinical implication

If 3DE is not available, we recommend using a 4CV single plane assessment because of its reduced foreshortening and LALS overestimation.

---

## [Decision Letter · Decision Letter 1]

31 Mar 2021

Reliability of left atrial strain reference values: A 3D echocardiographic study

PONE-D-20-30699R1

Dear Dr. Nabeshima,

We’re pleased to inform you that your manuscript has been judged scientifically suitable for publication and will be formally accepted for publication once it meets all outstanding technical requirements.

Kind regards,

Kumaradevan Punithakumar

Academic Editor

PLOS ONE

Additional Editor Comments (optional):

Reviewers' comments:

Reviewer's Responses to Questions

**Comments to the Author**

1. If the authors have adequately addressed your comments raised in a previous round of review and you feel that this manuscript is now acceptable for publication, you may indicate that here to bypass the “Comments to the Author” section, enter your conflict of interest statement in the “Confidential to Editor” section, and submit your "Accept" recommendation.

Reviewer #1: All comments have been addressed

2. Is the manuscript technically sound, and do the data support the conclusions?

Reviewer #1: Yes

3. Has the statistical analysis been performed appropriately and rigorously? 

Reviewer #1: Yes

4. Have the authors made all data underlying the findings in their manuscript fully available?

Reviewer #1: Yes

5. Is the manuscript presented in an intelligible fashion and written in standard English?

Reviewer #1: Yes

6. Review Comments to the Author

Reviewer #1: For the paper “Reliability of left atrial strain reference values: A 3D echocardiographic study”, the authors addressed my concerns in my review by performing more data analysis, giving more discussion, and adding the limitation. I recommend acceptance for publication.

7. PLOS authors have the option to publish the peer review history of their article (what does this mean?). If published, this will include your full peer review and any attached files.

Reviewer #1: **Yes: **Yijia Li

---

## [Editor Report · Acceptance letter]

5 Apr 2021

PONE-D-20-30699R1 

Reliability of left atrial strain reference values: A 3D echocardiographic study 

Dear Dr. Nabeshima:

I'm pleased to inform you that your manuscript has been deemed suitable for publication in PLOS ONE. Congratulations! Your manuscript is now with our production department. 

Kind regards, 

on behalf of

Professor Kumaradevan Punithakumar 

Academic Editor

PLOS ONE